# Multi-Agent Deep Reinforcement Learning for Multi-Robot Applications: A Survey

**DOI:** 10.3390/s23073625

**Published:** 2023-03-30

**Authors:** James Orr, Ayan Dutta

**Affiliations:** School of Computing, University of North Florida, Jacksonville, FL 32224, USA

**Keywords:** deep reinforcement learning, multi-robot systems, multi-agent learning, survey

## Abstract

Deep reinforcement learning has produced many success stories in recent years. Some example fields in which these successes have taken place include mathematics, games, health care, and robotics. In this paper, we are especially interested in multi-agent deep reinforcement learning, where multiple agents present in the environment not only learn from their own experiences but also from each other and its applications in multi-robot systems. In many real-world scenarios, one robot might not be enough to complete the given task on its own, and, therefore, we might need to deploy multiple robots who work together towards a common global objective of finishing the task. Although multi-agent deep reinforcement learning and its applications in multi-robot systems are of tremendous significance from theoretical and applied standpoints, the latest survey in this domain dates to 2004 albeit for traditional learning applications as deep reinforcement learning was not invented. We classify the reviewed papers in our survey primarily based on their multi-robot applications. Our survey also discusses a few challenges that the current research in this domain faces and provides a potential list of future applications involving multi-robot systems that can benefit from advances in multi-agent deep reinforcement learning.

## 1. Introduction

In a multi-robot application, several robots are usually deployed in the same environment [1,2,3]. Over time, they interact with each other via radio communication, for example, and coordinate to complete a task. Application areas include precision agriculture, space exploration, and ocean monitoring, among others. However, in all such real-world applications, many situations might arise that have not been thought of before deployment, and, therefore, the robots must need to plan online based on their past experiences. Reinforcement learning (RL) is one computing principle that we can use to tackle such dynamic and non-deterministic scenarios. Its primary foundation is trial and error—in a single-agent setting, the agent takes an action in a particular state of the environment, receives a corresponding reward, and transitions to a new state [4]. Over time, the agent learns which state–action pairs are worth re-experiencing based on the received rewards and which ones are not [5]. However, the number of state–action pairs becomes intractable, even for smallish computational problems. This has led to the technique known as deep reinforcement learning (DRL), where the expected utilities of the state–action pairs are approximated using deep neural networks [6]. Such deep networks can have hundreds of hidden layers [7]. Deep reinforcement learning has recently been used in finding a faster matrix multiplication solution [8], for drug discovery [9], to beat humans in Go [10], play Atari [6], and for routing in communication networks [11], among others. Robotics is no different—DRL has been used in applications ranging from path planning [12] and coverage [13] to locomotion learning [14] and manipulation [15].

Going one step further, if we introduce multiple agents to the environment, this increases the complexity [16]. Now, the agents not only need to learn from their own observations in the environment but also be mindful of other agents’ transitions. This essentially means that one agent’s reward may now be influenced by the actions of other agents, and this might lead to a non-stationary system. Although inherently more difficult, the use of multiple robots and, consequently, a multi-agent reinforcement learning framework for the robots is significant [17]. Such learning multi-robot systems (MRS) may be used for precision agriculture [18], underwater exploration [19], search and rescue [20], and space missions [21]. Robots’ onboard sensors play a significant role in such applications. For example, the state space of the robots might include the current discovered map of the environment, which could be created by the robots’ laser scanners [22]. The state might also include locations and velocities, for which the robot might need sensory information from GPS or an overhead camera [23]. Furthermore, vision systems, such as regular or multi-spectral cameras, can be used by the robots to observe the current state of the environment, and data collected by such cameras can be used for robot-to-robot coordination [24]. Therefore, designing deep reinforcement learning algorithms, potentially lightweight and sample-efficient, that will properly utilize such sensory information, is not only of interest to the artificial intelligence research community but to robotics as well. However, the last survey that reviewed the relevant multi-robot system application papers that use multi-agent reinforcement learning techniques was conducted by Yang and Gu in 2004 [17]. Note that the entire sub-field of *DRL was not invented until 2015* [6].

In this paper, we fill this significant void by reviewing and documenting relevant MRS papers that specifically use multi-agent deep reinforcement learning (MADRL). Since today’s robotic applications can have a large state space and, potentially, large action spaces, we believe that reviewing only the DRL-based approaches, and not the classic RL frameworks, is of interest to the relevant communities. The primary contribution of this article is that, to the best of our knowledge, this is the *only* study that surveys multi-robot applications via multi-agent deep reinforcement learning technologies. This survey provides a foundation for future researchers to build upon in order to develop state-of-the-art multi-robot solutions, for applications ranging from task allocation and swarm behavior modeling to path planning and object transportation. An illustration of this is shown in Figure 1.

We first provide a brief technical background and introduce terminologies necessary to understand some of the concepts and algorithms described in the reviewed papers (Section 2). In Section 3, we categorize the multi-robot applications into (1) coverage, (2) path planning, (3) swarm behavior, (4) task allocation, (5) information collection, (6) pursuit–evasion, (7) object transportation, and (8) construction. We identify and discuss a list of crucial challenges that, in our opinion, the current studies in the literature face in Section 4, and then, finally, we conclude.

## 2. Background

In this section, we provide technical backgrounds on relevant computing principles.

### 2.1. MDP and Q-Learning

Let *S* and *A* denote the set of all states and actions available to an agent. Let *R*: S×A→R denote a reward function that gives the agent a virtual reward for taking action a∈A in state s∈S. Let *T* denote the transition function. In a deterministic world, *T*: S×A→S, i.e., the actions of the agent is deterministic, whereas in a stochastic world, these actions might be probabilistic—*T*: S×A→prob(S). We can use a Markov Decision Process (MDP) to model such a stochastic environment, which is defined as a tuple 〈S,A,T,R〉. The objective is to find a (optimal) policy π: S→A that maximizes the expected cumulative reward. To give higher preference to the immediate rewards than to the future ones, we discount the future reward values. The sum of the discounted rewards is called value. Therefore, to solve an MDP, we will maximize the expected value (*V*) over all possible sequences of states. Thus, the expected utility in a state s∈S can be recursively defined as follows:(1)$$V\left(s\right)=R\left(s,a\right)+\gamma \max_{a^{\prime }\in A}\sum_{s^{\prime }}P\left(s^{\prime }|s,a\right)V\left(s^{\prime }\right)$$
The above is called the Bellman equation, where $P(s^{\prime }|s,a)$ is the probability of transitioning into $s^{\prime }$ from *s* by taking an action *a*. We can use value or policy iteration algorithms to solve an MDP.

However, in a situation where the *R* and *T* functions are unknown, the agent will have to try out different actions in every state to know which states are good and what action it should take in a particular state to maximize its utility. This leads to the idea of reinforcement learning (RL) where the agent will execute *a* in state *s* of the environment, and will receive a reward signal *R* from the environment as a result. Over time, the agent will learn the optimal policy based on this interaction between the agent and the environment [25]. An illustration is shown in Figure 2. In a *model-based* RL, the agent learns an empirical MDP by using estimated transition and reward functions. Note that these functions are approximated by interacting with the environment as mentioned earlier. Next, similar to an MDP, value or policy iteration algorithm can be employed to solve this empirical MDP model. In a *model-free* RL, the agent does not have access to *T* and *R*. This is true for numerous robotic applications in the real world. Therefore, most of the robotics papers we review in this survey use model-free RL techniques. This is also true for RL algorithms in general.

The goal of RL is to find a policy that maximizes the expected reward of the agent. Temporal difference learning is one of the most popular approach in model-free RL to learn learn the optimal utility values of each state. Q-learning is one such model-free RL technique, where the *Q*-value of a state–action pair (s,a) indicate the expected usefulness of that pair, which is updated as follows.
(2)$$Q\left(s,a\right)=\left(1-\alpha \right)Q\left(s,a\right)+\alpha (R\left(s,a\right)+\gamma \max_{a^{\prime }\in A}Q\left(s^{\prime },a^{\prime }\right))$$
α is the learning rate that weighs the new observations against the old. It is off-policy learning and converges to an optimal policy $\pi^{*}$ following
(3)$$\pi^{*}\left(s\right)=\arg\max_{a\in A}Q\left(s,a\right)$$

An excellent overview of classic RL applications in robotics can be found in [26]. Keeping track of Q-values for all possible state–action pairs in such an RL setting becomes infeasible with, for example, a million such combinations. In recent years, artificial neural networks have been used to approximate the optimal Q-values instead of storing the values in a table. This has given birth to the domain of *deep* reinforcement learning.

### 2.2. Multi-Agent Q-Learning

Assuming that the state space *S* is shared among *n* agents *N* and that there exists a common transition function *T*, an MDP for *N* is represented by the following tuple 〈N,S,A,O,T,R〉, where the joint action space is denoted by $A\leftarrow A_{1}\times A_{2}\cdots \times A_{n}$; the joint reward is denoted by $R\leftarrow R_{1}\times R_{2}\cdots \times R_{n}$; and O denotes the joint observation of the agents, $O\leftarrow O_{1}\times O_{2}\cdots \times O_{n}$. As there is more than one agent present, the action of one agent can potentially affect the reward and the consequent actions of the other agents. Therefore, the goal is to find a joint policy $\pi^{*}$. However, due to the non-stationary environment and, consequently, the removal of the Markov property, convergence cannot be guaranteed unlike the single-agent setting [27]. One of the earliest approaches to learning a joint policy for two competitive agents is due to Littman [28]. It was modeled as a zero-sum two-player stochastic game (SG). It is also known as Markov Game in game theory. In SG, the goal is to find the Nash equilibrium, assuming the *R* and *T* functions are known. In a Nash equilibrium, the agents (or the players) will not have any incentive to change their adopted strategies. We slightly abuse the notation here and denote the strategy of agent $N_{i}$ with $\pi_{i}$. Therefore, in a Nash equilibrium, the following is true
(4)$$V_{i}^{\pi_{i}^{*},\pi_{-i}^{*}}\left(s\right)\geq V_{i}^{\pi_{i}^{,}\pi_{-i}^{*}}\left(s\right),\forall \pi_{i}$$
where V(s) denotes the value of state s∈S to the *i*-th agent and $\pi_{-i}$ is the strategy of the other players. Here, we assume the agents to be rational, and, therefore, all the agents always follow their optimal strategies. This general SG setting can now be used to solve multi-agent reinforcement learning (MARL) problems.

In a cooperative setting, the agents have a common goal in mind. Most of the studies in the robotics literature that use MARL use such a cooperative setting. In this case, the agents have the same reward function, *R*. Given this, all the agents in *N* will have the same value function, and, consequently, the same Q-function. The Nash equilibrium will be the optimal solution for this problem. Two main types of learning frameworks are prevalent—independent and joint learners. In an independent learning scenario, each agent ignores the presence of other agents in the environment and considers their influence as noise. The biggest advantage is that each agent/robot can implement its own RL algorithm and there is no need for coordination and, consequently, a joint policy calculation [16,27]. Independent classic Q-learners have shown promising results in AI [29,30], as well as in robotics [31,32]. On the other hand, the joint learners aim to learn the joint optimal policy from O and A. Typically, an explicit coordination, potentially via communication in an MRS, is in place and the agents learn a better joint policy compared to the independent learners [16,27]. However, the complexity increases exponentially with the number of agents causing these not to scale very well. The joint Q-learning algorithms are also popular in robotics [24,33,34], as well as in general AI [28,35]. A comprehensive survey for MARL techniques can be found in [27,36]. The authors in [36] also discuss the application domains for MARL, which includes multi-robot teams. A specific relevant example that is discussed is multi-robot object transportation.

### 2.3. (Multi-Agent) Deep Q-Learning

As the state and the action spaces increase in size, maintaining a table for the Q-values for all possible state–action pairs might be infeasible. To tackle this challenge, Mnih et al. [6] have proposed a neural network-based approach to approximate the Q-values directly from the sensory inputs. This has given birth of ‘deep’ Q-learning, as the Q-values of the state–action pairs are updated using a deep neural network.

#### 2.3.1. Q-Networks

In their seminal paper, Mnih et al. [6] have proposed DQN—a convolutional neural network (CNN) to approximate the Q-values for a single agent. This is called the Q-network, which is parameterized by θ. The current state $s_{t}$ is passed as an input to the network that outputs the Q-values for all the possible actions. An action is chosen next based on the highest Q-value, i.e.,
(5)$$a^{*}=\arg\max_{a\in A}Q\left(s_{t},a\right)$$

To ensure that the agent explores the state space enough, $a^{*}$ is chosen with probability ϵ and the agent takes a random action with (1−ϵ) probability. Due to this action, the state transitions to $s_{t+1}$. To avoid instability, a *target* network is maintained—it is identical to the Q network, but the parameter set θ is periodically copied to the parameters of this target network, $\theta^{-}$. The state transitions are maintained in an experience replay buffer D. Mini-batches from D are selected and target Q-values are predicted. θ is regressed toward the target values by finding the gradient descent of the following temporal loss function
(6)$$L=E[{\left(y_{t}-Q\left(s_{t},a_{t}\right)\right)}^{2}]$$
(7)$$y_{t}=R+\gamma Q\left(s_{t+1},\arg\max Q\left(s_{t+1},a_{t+1}\right)\right)$$

One of the most popular extensions of DQN is Double DQN (DDQN) [37], which reduces overestimation in Q-learning. DDQN uses the Q-network for action selection following the ϵ-greedy policy, as mentioned above, but uses the target network for the evaluation of the state–action values. DQN and DDQN are extremely popular in robotics [13,38,39,40,41]. A visual working procedure of the generic DQN algorithm is presented in Figure 3.

#### 2.3.2. Policy Optimization Techniques

In policy optimization methods, the neural network outputs the probability distribution of these actions instead of outputting the Q-values of the available actions. Instead of using something like the ϵ-greedy strategy to derive a policy from the Q-values, the actions with higher probability outputs from the network will have higher chances of being selected. Let us denote a θ-parameterized policy by $\pi_{\theta }$. The objective is to maximize the cumulative discounted reward
(8)$$J\left(\theta \right)=E\left[R|\pi_{\theta }\right]$$
where R is the finite-horizon discounted cumulative reward. By optimizing the parameter set θ, e.g., by following the gradient of the policy, we aim to maximize the expected reward. Similar to the Q-networks, the learning happens in episodes. In general, the parameters in episode i+1, $\theta_{i+1}$, will be an optimized version of $\theta_{i}$ as the following standard gradient ascent formula
(9)$$\theta_{i+1}=\theta_{i}+\alpha \frac{\delta J(\theta_{i})}{\delta \theta_{i}}.$$

In the vanilla form, similar to the Q-networks, the mean square error between the value of the policy (usually approximated using a neural network) and the reward-to-go (i.e., the sum of rewards received after every state transition so far) is calculated and the approximate value function parameters are regressed. Some of the popular policy optimization techniques include Deep Deterministic Policy Gradient (DDPG) [42], Proximal Policy Optimization (PPO) [43], Trust Region Policy Optimization (TRPO) [44], and Asynchronous Advantage Actor–Critic (A3C) [45], among others. Among these, DDPG is one of the most widely used for multi-robot applications [46,47,48,49,50]. It learns a Q-function similar to DQN and uses that to learn a policy. The policy DDPG learns is deterministic and the objective of this is to find actions that maximize the Q-values. As the action space *A* is assumed to be continuous, the Q-function is differentiable. To optimize θ and update the policy, we perform one-step policy ascent as follows:(10)$$\max_{\theta }E_{s\in D}\left[Q\left(s,\pi_{\theta }\left(s\right)\right)\right].$$

DDPG uses a sophisticated technique called actor–critic to achieve the successful combination of these two types of deep Q-learning. The actor essentially represents the policy and the critic represents the value network, respectively. The actor is updated towards the target and the critic is regressed by minimizing the error with the target [51]. The difference between the expected state value and the Q-value for an action *a* is called the *advantage*. One of the most popular algorithms that uses such an actor–critic framework is A3C [45]. In this algorithm, parallel actors explore the state space via different trajectories making the algorithm asynchronous; therefore, it does not require maintaining an experience replay. Another popular algorithm in the multi-robot domain is PPO, potentially because of its relatively simple implementation [43]. PPO-clip and PPO-penalty are its two primary variants that are used in robotics [52,53,54,55,56,57].

#### 2.3.3. Extensions to Multi-Agent

As described earlier, in independent learning frameworks, any of the previously mentioned deep RL techniques, such as DQN, DDPG, A3C, or PPO, can be implemented on each agent. Note that no coordination mechanism is needed to be implemented for this [16,27,58].

For multi-agent DQN, a common experience memory can be used, which will combine the transitions of all the agents, and, consequently, they will learn from their global experiences while virtually emulating a stationary environment. Each agent can have its own network that will lead it to take an action from its Q-values [59]. Yang et al. [60] have proposed a mean field Q-learning algorithm for large-scale multi-agent learning applications. A mean-field formulation essentially brings down the complexity of an *n*-agent learning problem to a 2-agent learning problem by creating a virtual mean agent from the other (n−1) agents in the environment. In [61], the authors have introduced the multi-agent extension of DDPG (MADDPG). Here, the actor remains decentralized, but the critic is centralized. Therefore, the critic needs information on the actions, observations, and target policies of all of the agents to evaluate the quality of the joint actions. Figure 4 shows an illustration of this process. Yu et al. [62] have proposed a multi-agent extension of PPO in cooperative settings (MAPPO). Similar to MADDPG, it uses centralized training with a decentralized execution strategy.

Another approach to extending a single-agent DRL algorithm to a multi-agent setting is to model it as a centralized RL, where all the information from agents is input together. This might create an infeasibly large state and action space for the joint agent. To alleviate this, researchers have looked into how to find each agent’s contribution to the joint reward. This is named Value Function Factorization. VDN [63] is one such algorithm for cooperative settings where the joint Q-value is the addition of the local Q-values of the agents. A summary of the main types of RL algorithms used in multi-robot applications is presented in Table 1. The reader is referred to [64,65] for recent comprehensive surveys on state-of-the-art MADRL techniques and challenges. Furthermore, Oroojlooy and Hajinezhad [66] have recently published a survey paper reviewing the state-of-the-art MADRL algorithms specifically for cooperative multi-agent systems. As in most of the scenarios, the robots in an MRS work together towards solving a common problem, we believe that the survey in [66] would be a valuable asset for the robotics community.

## 3. Multi-Robot System Applications of Multi-Agent Deep Reinforcement Learning

A summary of the discussed multi-robot applications is presented in Figure 5.

### 3.1. Coverage and Exploration

The goal of an MRS in a coverage path planning (CPP) application is that every point in the environment is visited by at least one robot while some constraints are satisfied (e.g., no collision among the robots) and user-defined criteria are optimized (e.g., minimizing the travel time) [145]. CPP is one of the most popular topics in robotics. For multi-robot coverage, several popular algorithms exist even with performance guarantees and worst-case time bounds [146,147,148,149]. In exploration, however, the objective might not be the same as the multi-robot CPP problem. It is assumed that the sensor radius r>0, and, therefore, the robots do not need to visit all the points on the plane. For example, the robots might be equipped with magnetic, acoustic, or infrared sensors in ground and aerial applications whereas a group of underwater vehicles might be equipped with water temperature and current measuring sensors. The robots will need GPS for outdoor localization. Such exploration can be used for mapping and searching applications among others [150,151,152]. Constraints such as maintaining wireless connectivity for robots with limited communication ranges might be present [153]. Inter-robot communication can be achieved via ZigBee or Wi-Fi. An example is shown in Figure 6.

Mou et al. [68] studied area coverage problems and proposed deep reinforcement learning for UAV swarms to efficiently cover irregular three-dimensional terrain. The basis of their UAV swarm structure is with the leader and the follower UAVs. The authors implement an observation history model based on convolutional neural networks and a mean embedding method to address limited communication. Li et al. [69] proposed the use of DDQN to train individual agents in a simulated grid-world environment. Then during the decision-making stage, where previously trained agents are placed in a test environment, the authors use their proposed multi-robot deduction method, which has foundations in Monte Carlo Tree Search. Zhou et al. [154] have developed a multi-robot coverage path planning mechanism that incorporates four different modules: (1) a map module, (2) a communication module, (3) a motion control module, and (4) a path generation module. They implement an actor–critic framework and natural gradient for updating the network. Up to three robots have been used in a simulation for testing the proposed coverage technique in a grid world. The two cornerstones of the study by Hu et al. [155] are (1) Voronoi partitioning-based area assignment to the robots and (2) the proposed DDPG-based DRL technique for the robots to have a collision-avoidance policy and evade objects in the field. The control of the robots is provided by the underlying neural network. The authors use a Prioritised Experience Replay (PER) [156] to store human demonstrations. The simulation was performed within Gazebo [157] and three Turtlebot3 Waffle Pi mobile robots were used to explore an unknown room during validation. Bromo [53], in his thesis, used MADRL on a team of UAVs using a modified version of PPO to map an area. During training, the policy function is shared among the robots and updated based on their current paths.

For multi-UAV coverage, Tolstaya et al. [110] use graph neural networks (GNNs) [158] as a way for the robots to learn the environment through the abstractions of nodes and edges. GNNs have been successfully used in various coordination problems for multi-robot systems, and more recently, Graph Convolution Networks (GCNs) have been used [158]. The authors in this paper use “behavior cloning” as a heuristic to train the GNN on robots’ previous experiences. In order for the individual UAVs to learn information distant from their position, they use up to 19 graph operation layers. PPO is the base DRL algorithm in this paper. Aydemir and Cetin [159] proposed a distributed system for the multi-UAV coverage in partially observable environments using DRL. Only the nearby robots share their state information and observations with each other. Blumenkamp et al. [111] developed a framework for decentralized coordination of an MRS. The RL aspect of their system uses GNNs and PPO. The agents train and develop a policy within a simulated environment and then the physical implementation of the policy with the robots occurs in a test environment. The authors also compare centralized control and communication levels to decentralized decision-making.

Similarly, Zhang et al. [160] have also proposed to employ graph neural networks for multi-robot exploration. The authors emphasize the “coarse-to-fine” exploration method of the robots, where the graph representation of the state space to be explored is explored in “hops” of greater detail. Simulation experiments involved up to 100 robots. Exploration can also be used for searching for a target asset. Liu et al. [84] have proposed a novel algorithm for cooperative search missions with a group of unmanned surface vehicles. Their algorithm makes use of two modules based on a divide-and-conquer architecture: an environmental sense module that utilizes sensing information and a policy module that is responsible for the optimal policy of the robots. Gao and Zhang [161] study a cooperative search problem while using MADRL as the solution method. The authors use independent learners on the robots to find the Nash equilibrium solution with the incomplete information available to the robots. Setyawan et al. [101] also use MADRL for multi-robot search and exploration. Unlike the previously mentioned studies, the authors have adopted a hierarchical RL approach, where they break down an abstraction of the global problem space into smaller sub-problem levels in order for the robot system to more efficiently learn in an actor–critic style. The lowest level in this order decides the robots’ motor actions in the field. Sheng et al. [162] propose a novel probability density factorized multi-agent DRL method for solving the multi-robot reliable search problem. According to this study, when each robot follows its own policy to maximize its own reliability metric (e.g., probability of finding the target), the global reliability metric is also maximized. The authors implement the proposed technique on multiple simulated search environments including offices and museums, as well as on real robots. Another study in a similar application domain is done by Xia et al. [127]. Specifically, the authors have used MADRL for the multi-agent multi-target hunting problem. The authors make use of a feature embedding block to extract features from the agents’ observations. The neural network architecture uses fully connected layers and a Gated Recurrent Unit (GRU) [163]. Simulation experiments included up to 24 robots and 12 targets. Caccavale et al. [96] proposed a DRL framework for a multi-robot system to clean and sanitize a railway station by coordinating the robots’ efforts for maximum coverage. Their approach is decentralized where each robot runs its own CNN and the foundation of their technique is DQN. Note that the robots learn to cooperate online while taking the presence of the passengers into account.

Not only with the ground and aerial vehicles, but MADRL has also been used for ocean monitoring with a team of floating buoys as well. Kouzehgar et al. [105] proposed two area coverage approaches for such monitoring: (1) swarm-based (i.e., the robots follow simple swarming rules [164]) and (2) coverage-range-based (i.e., the robots with fixed sensing radius). The swarm-based model was trained using MADDPG and the latter model MARL was trained using a modified (consisting of eliminating reward sharing, collective reward, sensing their own share of the reward function, and independence based on individual reward) MADDPG algorithm.

Communication is one of the most important methods of coordination among a group of robots. More often than not, when and with whom the communication will happen is pre-defined. However, if the robots are non-cooperative, such an assumption does not work. Blumenkamp and Prorok [118] propose a learning model based on reinforcement learning that allows individual, potentially non-cooperative, agents to manipulate communication policies while the robots share a differentiable communication channel. The authors use GNN with PPO in their method. The proposed technique has also been successfully employed for multi-robot path planning. Along a similar path, Liang et al. [165] proposed the use of DRL to learn a high-level communication strategy. The authors presume the environment to be partially observable and they take a hierarchical learning approach. The implemented application is a cooperative patrolling field with moving targets. Meng and Kan [102] also put multi-robot communication at the forefront of their study while tackling the coverage problem. The goal of the robots has to cover an entire environment while maintaining connectivity in the team, e.g., via a tree topology. The authors use a modified version of MADDPG to solve the stated problem.

MADRL has also been used for sensor coverage, alongside area coverage [166]. In sensor-based coverage, the objective is to cover all the points in an environment with a sensor footprint. An example of this is communication coverage, where the goal of a team of UAVs is to provide Wi-Fi access to all the locations in a particular region. This might be extremely valuable after losing communication in a natural disaster, for example. The authors in [167] presented a solution for UAV coverage using mean field games [168]. This study was targeted toward UAVs that provide network coverage when network availability is down due to natural disasters. The authors constructed the Hamilton–Jacobi–Bellman [169] and Fokker–Planck–Kolmogorov [170] equations via mean field games. Their proposed neural network-based learning method is a modification of TRPO [44] and named mean-field trust region policy optimization (MFTRPO). Liu et al. [104] proposed a coverage method to have a system of UAVs cover an area and provide communication connectivity while maintaining energy efficiency and fairness of coverage. The authors utilize an actor–critic-based DDPG algorithm. Simulation experiments were carried out with up to 10 UAVs. Similar to these, Nemer et al. [171] proposed a DDPG-based MADRL framework for multi-UAV systems to provide better coverage, efficiency, and fairness for network coverage of an area. One of the key differentiating factors of this paper is that the authors also model energy-efficient controls of the UAVs to reduce the overall energy consumption by them during the mission. For a similar communication coverage application, Liu et al. [172] proposed that the UAVs have their own actor-critic networks for a fully-distributed control framework to maximize temporal mean coverage reward.

### 3.2. Path Planning and Navigation

In multi-robot path planning (or path finding), each robot is given a unique start and a goal location. Their objective is to plan a set of joint paths from the start to the goal, such that some pre-defined criteria, such as time and/or distance, are optimized and the robots avoid colliding with each other while following the paths. An illustration is presented in Figure 7. Planning such paths optimally has been proven to be NP-complete [173]. Like $A^{*}$ [174], which is used for single-agent path planning in a discreet space, $M^{*}$ [175] can be used for an MRS. Unfortunately, $M^{*}$ lacks scalability. There exist numerous heuristic solutions for such multi-robot planning that scale well [176,177,178,179]. Overhead cameras and GPS can be used to localize the robots in indoor and outdoor applications, respectively. In GPS and communication-denied environments, vision systems can be used as a proxy [180]. Recently, researchers have started looking into deep reinforcement learning solutions to solve this notoriously difficult problem.

One of the most popular works that use MADRL for collision avoidance is due to Long et al. [22]. They propose a decentralized method using PPO while using CNNs to train the robots, which use their onboard sensors to detect obstacles. Up to 100 robots were trained and tested via simulation. Lin et al. [109] proposed a novel approach for centralized training and decentralized execution for a team of robots that need to concurrently reach a destination while avoiding objects in the environment. The authors implement their method using CNNs and PPO as well. The learned policy maps LiDAR measurements to the controls of the robots. Bae et al. [72] also use CNNs to train multiple robots to plan paths. The environment is treated as an image where the CNN extracts the features from the environment, and the robots share the network parameters.

Fan et al. [107] have proposed a DRL model technique using the policy gradient method to train the robots to avoid collisions with each other while navigating in an environment. The authors use LiDAR data for training, and, during testing, this drives the decision-making process to avoid collisions. The authors then transfer the learned policy to physical robots for real-world feasibility testing. The simulation included up to 100 robots with the objective of avoiding collisions with each other, static objects, and, finally, pedestrians. It builds on their previous work from 2018 [131]. Wang et al. [181] also use CNNs for multi-robot collision avoidance and coordination. The authors also use a recurrent module, namely Long Short Term Memory (LSTM) [182] to memorize the actions of the robots to smooth the trajectories. The authors have shown that the combined use of CNN and LSTM can produce smoother paths for the robots in a continuous domain.

Yang et al. [71] use a priori knowledge to augment the DDQN algorithm to improve the learning efficiency in multi-robot path planning. To avoid random exploration at the beginning of the learning process, the authors have used $A^{*}$ [174] paths for single robots in static environments. This provides better preliminary Q-values to the networks, and, thus, the overall learning process converges relatively quickly. Wang and Deng [39] propose a novel neural network structure for task assignment and path planning where one network processes a top–down view of the environment and another network processes the first-person view of the robot. The foundation of the algorithm is also based on DQN. Na et al. [49] have used MADRL for collision avoidance among autonomous vehicles via modeling virtual pheromones inspired by nature. The authors also used a similar pheromone-based technique, along with a modified version of PPO in [55] for the same objective. Ourari et al. [123] also used a biologically-inspired method (specifically from the behavior of flocks of starlings) for multi-robot collision avoidance while a DRL method, namely PPO, is at its foundation. Their method is executed in a distributed manner and each robot incorporates information from *k*-nearest neighbors.

For multi-robot target assignment and navigation, Han, Chen, and Hao [117] proposed to train the policy in a simulated environment using randomization to decrease the performance transfer from simulation to the real world. The architecture they developed utilized communication amongst the robots to share experiences. They also developed a training algorithm for navigation policy, target allocation, and collision avoidance. It uses PPO as a foundation. Moon et al. [38] used MADRL for the coordination of multiple UAVs that track first responders in an emergency response situation. One of the key ideas behind their method is the inclusion of the Cramér–Rao lower bound into the learning process. The intent of the authors was to use the DRL-based UAV control algorithm to accurately track the target(s) of the UAV system. They used DDQN as their foundation technique.

Marchesini and Farinelli [74] extended their work in [75] by incorporating an Evolutionary Policy Search (EPS) for multi-robot navigation. It had two main components: navigation (reaching a target) and avoiding collisions. They extended their prior work [75] (using DDQN and LSTM at its core) by incorporating the EPS, which integrated randomization and genetic learning into the MARL technique to enhance the ability for the policy to explore and help the robots learn to navigate better.

Lin et al. [112] developed a novel deep reinforcement learning approach for coordinating the movements of an MRS such that the geometric center of the robots reached a target destination while maintaining a connected communication graph throughout the mission. Similarly, Li et al. [183] proposed a DRL method for multi-robot navigation while maintaining connectivity among the robots. The presented technique used constrained policy optimization [184] and behavior cloning. Real-world experiments with five ground robots show the efficacy of the proposed method. Maintaining such connectivity has previously been studied in an information collection application [185] applied to precision agriculture, albeit from a combinatorial optimization perspective [18].

On the other hand, Huang et al. [167] proposed a deep Q-learning method for maintaining connectivity between leader and follower robots. Interestingly, the authors do not use CNNs, instead, they rely only on dense fully connected layers in their network. Similar to these, Challita et al. [167] developed a novel DRL framework for UAVs to learn an optimal joint path while maintaining cellular connectivity. Their main contribution is founded in game theory. The authors used an Echo State Network (ESN), a type of recurrent neural network. In a similar setting, the authors’ other work [186] studied minimizing interference from the cellular network using MADRL. Wang et al. [187] proposed to incorporate environmental spatiotemporal information. The proposed method used a global path planning algorithm with reinforcement learning at the local level via DDQN combined with an LSTM module. Choi et al. [92] also used a recurrent module, namely GRU along with CNN for the multi-agent path planning problem in an autonomous warehouse setting. The base of their work was the popular QMIX [188] algorithm, a form of value function factorization algorithm similar to VDN [63]. Another study of multi-robot path planning for warehouse production scenarios was carried out by Li and Guo [128]. They proposed a supervised DRL approach efficient path planning and collision avoidance. More specifically, using imitation learning and PPO, Li and Guo aimed to increase the learning performance of the vehicles in object transportation tasks.

Yao et al. [115] developed a map-based deep reinforcement learning approach for multi-robot collision avoidance, where the robots do not communicate with one another for coordination. The authors used an egocentric map as the basis of information that the robots use to avoid collisions. Three robots have been used for real-world implementations. Similar to this, Chen et al. [189] also did not rely on inter-robot communication for multi-robot coordinated path planning and collision avoidance while also navigating around pedestrians. Chen et al. [94]’s study on multi-robot path planning also considered non-communicating and decentralized agents using DDQN. Simulation experiments involved up to 96 robots.

Tan et al. [116] have developed a novel algorithm, called DeepMNavigate that uses local and global map information, PPO, and CNNs for navigation and collision avoidance learning. Their algorithm also makes use of multi-staged training for robots. Simulation experiments involved up to 90 robots. Chen et al. [190] proposed a method of using DRL in order for robots to learn human social patterns to better avoid collisions. As human behaviors are difficult to model mathematically, the authors noted that social rules usually emerge from local interactions, which drives the formulation of the problem. Chen et al. [87] proposed a novel DRL framework using hot-supervised contrastive loss (via supervised contrastive learning) combined with DRL loss for pathfinding. The robots do not use communication. They also incorporated a self-attention mechanism in the training. Their network structure used CNNs with DQN while up to 64 agents have been used for testing the approach in simulation. Navigation control using MADRL was also studied in [88], where the authors showed that the robots could recover from reaching a dead end. Alon and Zhou [135] have proposed a multi-critic architecture that also included multiple value networks. Path planning is also important in delivering products to the correct destinations. Ding et al. [91] have proposed a DQN-based MADRL technique for this specific application while combining it with a classic search technique, namely the Warshall–Floyd algorithm.

Transfer learning and federated deep learning have also been used for multi-robot path planning. In transfer learning, the assumption that the training and the testing data come from the same domain does not need to hold, which makes it attractive in many real-world scenarios, including robotics [191]. The objective here is to *transfer* the learning from one or more source domains to a potentially different target domain. Wen et al. [133] developed two novel reinforcement learning frameworks that extend the PPO algorithm and incorporate transfer learning via meta-learning for path planning. The robots learn policies in the source environments and obtain their policies following the proposed training algorithm. Next, this learning is then transferred to target environments, which might have more complex obstacle configurations. This increases the efficiency of finding the solutions in the target environments. The authors used LSTM in their neural network for memorizing the history of robot actions. In federated deep learning, training data might still be limited similar to the transfer learning applications. In this case, each agent has its own training data instead of using data shared by a central observer [192]. For example, each robot might have access to a portion of the environment, and they are not allowed to share the local images with each other, where the objective is still to train a high-quality global model. Luo et al. [193] have employed such a federated deep RL technique for multi-robot communication. The authors, in this paper, avoid blockages in communication signals due to large obstacles while avoiding inter-robot collisions. It has been shown that the proposed semi-distributed optimization technique is 86% more efficient than a central RL technique. Another federated learning-based path planning technique can be found in [130]. To reduce the volume of exchanged data between a central server and an individual robot, the proposed technique only shares the weights and biases of the networks from each agent. This might be significant in scenarios where the communication bandwidth is limited. The authors show that the presented technique in their paper offers higher robustness than a centralized training model.

PRIMAL is a multi-agent path-finding framework that uses MADRL and is proposed by Sartoretti et al. [194]. PRIMAL used the A3C [45] algorithm and an LSTM module. It also makes use of imitation learning whereby each agent can be given a copy of the centrally trained policy by an expert [195]. One of the highlights of this paper is that the proposed technique could scale up to 1024 robots albeit in simulation. PRIMAL${}_{2}$ [196] is the advanced version of PRIMAL and was proposed by Damani et al. in 2021. It also uses A3C as its predecessor, offers real-time path re-planning, and scales up to 2048 robots—double that which PRIMAL could do.

Curriculum learning [197] has also been used for multi-robot path planning in [198], where the path planning is modeled as a lesson, going from easy to hard difficulty levels. An end-to-end MADRL system for multi-UAV collision avoidance using PPO has been proposed by Wang et al. [57]. Asayesh et al. [137] proposed a novel module for safety control of a system of robots to avoid collisions. The authors use LSTM and a Variational Auto-Encoder [199]. Li [200] has proposed using a lightweight decentralized learning framework for multi-agent collision avoidance by using only a two-layer neural network. Thumiger and Deghat [56] used PPO with an LSTM module for multi-UAV decentralized collision avoidance. Along the same line, Han et al. [54] used GRUs and their proposed reward function used reciprocal velocity obstacle for distributed collision avoidance.

For collaborative motion planning with multiple manipulators, Zhao et al. [108] proposed a PPO-based technique. The manipulators learned from their own experiences, and then, a common policy was updated while the arms continued to learn from individual experiences. This created differences in accuracy or actuator ability among the manipulators. Similarly, Gu et al. [50] proposed a method for asynchronous training of manipulator arms using DDPG and Normalized Advantage Function (NAF). Real-world experiments were carried out with two manipulators. Prianto et al. [143] proposed the use of the Soft Actor–Critic (SAC) algorithm [14] due to its efficiency in exploring large state spaces for path planning with a multi-arm manipulator system, i.e., each arm has its own unique start and goal configurations. Unlike the previous works in this domain, the authors used Hindsight Experience Replay (HER) [201] for sample-efficient training. On the other hand, Cao et al. [144] proposed a DRL framework for a multi-arm manipulator to track trajectories. Similarly to [143], Cao et al. also used SAC as their base algorithm. The main distinguishing factor of this study is that the multiple manipulator arms were capturing a non-cooperative object. Results show that the dual-arm manipulator can capture a rotating object in space with variable rotating speeds. An illustration of such a dual-arm manipulation application is shown in Figure 8.

Everett et al. [202] have proposed to use LSTM and extend their previous DRL algorithm [189] for multi-robot path planning to enhance the ability of the robots to avoid collisions. Semnani et al. [203] proposed an extension of the work proposed in [202] by using a new reward function for multi-agent motion planning in three-dimensional dense spaces. They used a hybrid control framework by combining DRL and force-based motion planning. Khan et al. [136] have proposed using GCN and a DRL algorithm called Graph Policy Gradients [134] for unlabeled motion planning of a system of robots. The multi-robot system must find the goal assignments while optimizing their trajectories.

Song et al. [90] designed a new actor–critic algorithm and a method for extracting the state features via a local-and-global attention module for a more robust MADRL method with an increasing number of agents present in the environment. The simulated experiments used dynamic environments with simulated pedestrians. Zhang et al. [204] proposed a method for using a place-timed Petri net and DRL for the multi-vehicle path planning problem. They used a curriculum-based DRL model. Huang et al. [205] proposed a vision-based decentralized policy for path planning. The authors use Soft Actor–Critic with auto encoders [206] as their deep RL technique for training a multi-UAV system. The 3D images captured by the UAVs and their inertial measurement values were used as inputs, whereas the control commands were rejected by the neural network. Simulation experiments with up to 14 UAVs were performed within the Airsim simulator. Jeon et al. [207] proposed to use MADRL to improve the energy efficiency of coordinating multiple UAVs within a logistic delivery service. The authors show that their model performs better in terms of consumed energy while delivering similar numbers of goods.

MADRL has also found its way into coordinating multiple autonomous vehicles. The authors in [208] provide a solution to the “double merge” scenario for autonomous driving cars that consists of three primary contributions in this field: (1) the variance of the gradient estimate can be minimized without Markovian assumptions, (2) trajectory planning with hard constraints to maintain the safety of the maneuver [209], and (3) introduction of a hierarchical temporal abstraction [25] that they call an “Option Graph” to reduce the effective horizon which ultimately reduces the variance of the gradient estimation [210,211]. Similar to this, Liang et al. [212] have modeled the cooperative lane changing problem among autonomous cars as a multi-agent cooperation problem and solved it via MADRL. Specifically, the authors have used a hierarchical DRL method that breaks down the problem into “high-level option selection” and “low-level control” of the agent. Real-world experiments were performed using a robotic test track with four robots, where two of them performed the cooperative lane change.

Finally, Sivanathan et al. [119] proposed a decentralized motion planning framework and a Unity-based simulator specifically for a multi-robot system that uses DRL. The simulator can handle both independent learners and common policies. The simulator was tested with up to four cooperative non-holonomic robots that shared limited information. PPO was used as the base algorithm to train the policies.

### 3.3. Swarm Behavior Modeling

Navigation of a swarm of robots through a complex environment is one of the most researched topics in swarm robotics. To have a stable formation, each robot should be aware of the positions of the nearby robots. A swarm consisting of miniature robots might not have a sophisticated set of sensors available. For example, a compass can be used to know the heading of the robot. Additionally, range and bearing sensors can also be available [213,214]. Infrared sensors can be used for communication in such a swarm system [215]. Inspired by swarms of birds or schools of fish, robots usually follow three simple rules to maintain such formations: cohesion, collision avoidance, and velocity alignment [164]. It is no surprise that multi-agent deep reinforcement learning techniques have been extensively employed to mimic such swarm behaviors and solve similar problems. An illustration of forming a circle with a swarm of five e-puck robots is presented in Figure 9.

Zhu et al. [216] proposed a novel algorithm for multi-robot flocking. The algorithm builds on MADDPG and uses PER. Results from three robots show that the proposed algorithm improves over the standard MADDPG. Similarly, Salimi and Pasquier [106] have proposed the use of DDPG with centralized training and a decentralized execution mechanism to train the flocking policy for a system of UAVs. Such flocking with UAVs might be challenging due to complex kinematics. The authors show that the UAVs reach the flocking formation using a leader–follower technique without any parameter tuning. Lan et al. [217] developed a control scheme for the cooperative behavior of a swarm. The basis of their control scheme is pulled from joint multi-agent reinforcement learning theory, where the robots not only share state information, but also a performance index designed by the authors. Notably, the convergence of the policy and the value networks is theoretically guaranteed. Following the above-mentioned works, Kheawkhem and Khuankrue [99] also proposed using MADDPG to solve the multi-agent flocking control problem.

Qiu et al. [100] used MADRL to improve sample efficiency, reduce overfitting, and allow better performance, even when agents had little or “bad” sample data in a flocking application. The main idea was to train a swarm offline with demonstration data for pre-training. The presented method is based on MADDPG. Not only for coverage as described earlier, but GNNs are also popular in general for coordination in a swarm system, especially in spatial domains. For example, Kortvelesy and Prorok [218] developed a framework, called ModGNN, which aimed to provide a generalized, neural network framework, that can be applied to varying multi-robot applications. The architecture is modular in nature. They tested the framework for a UAV flocking application with 32 simulated robots.

Yan et al. [219] studied flocking in a swarm of fixed-wing UAVs operating in a continuous space. Similar studies on flocking can also be found in more recent papers from these authors [83,220]. Similar to Yan et al.’s body of work, Wang et al. [142] proposed a TD3-based [221] solution for a similar application—flocking with fixed-wing UAVs where the authors test the method with up to 30 simulated UAVs. Not strictly for swarms, Lyu et al. [47] addressed the multi-agent flocking control problem specifically for a multi-vehicle system using DDPG with centralized training and decentralized execution. Notably, the authors take connectivity preservation into account while designing their reward function—the maximum distance could not go beyond the communication range and the minimum distance was kept at $d_{s}$, a physically safe distance between two vehicles. Interestingly, the mission waypoints are pre-defined in this paper. Bezcioglu et al. [48] also study flocking in a swarm system using DDPG and CNN, and tested it with up to 100 robots. The authors have used bio-inspired self-organizing dynamics for the joint motion of the robots.

Wang et al. [113] used MADRL to organize a swarm in specific patterns using auto-encoders [222] to learn compressed versions of the states and they tested the presented solution with up to 20 robots. Li et al. [46] proposed using a policy gradient method, namely MADDPG, with an actor–critic structure for circle formation control with a swarm of quad-rotors. Although circle formation is a popular application [223,224,225,226], this is one of the few studies that employed MADRL techniques. Sadhukhan and Selmic [121] have used PPO in order to train a multi-robot system to navigate through narrow spaces and reform into a designated formation. They used two reward schemes (one individual to the agents and one depending on the contributions to the team) and the system was centrally trained. In [125], Sadhukhan and Selmic extended their prior works by proposing a bearing-based reward function for training the swarm system, which utilizes a single policy shared among the robots.

Chen et al. [97] have developed an improved DDPG to enhance the ability of a robot to learn human intuition-style navigation without using a map. Furthermore, they create a parallel version of DDPG to extend their algorithm to a multi-robot application. Thereby, providing the robots with a method of sharing information/experiences in order to maintain formation, navigate an indoor environment, and avoid collisions. Qamar et al. [138] proposed novel reward functions and an island policy-based optimization framework for multiple target tracking using a swarm system. Along a similar line, Ma et al. [98] developed a DDPG-based algorithm for multi-robot formation control around a target, particularly in a circle around a designated object. The algorithm allows the robots to independently control their actions using local teammates’ information.

Recently, Zhang et al. [124] have also proposed a target encirclement solution that uses a decentralized DRL technique. The main contribution of their work is the use of three relational graphs among the robots and other entities in the system designed using a graph attention network [227]. In their simulation experiments, the authors use six robots encircling two targets. Similarly, Khan et al. [134] have used a graph representation of the robot formation and proposed using graph convolutional neural networks [158,228,229] to extract features, i.e., local features of robot formations, for policy learning. Simulation policies were trained on three robots and then the policy is transferred to over 100 robots for testing. The robots are initialized to certain positions and are to form a specific formation while reaching an end goal.

Zhou et al. [230] recognized the problem of computational complexity with existing MADRL methods for multi-UAV multi-target tracking while proposing a decentralized solution. Their proposed solution has its root in the reciprocal altruism mechanism of cooperation theory [231]. The experience replay is shared among the UAVs in this work. Zhou et al. [139] also study target tracking with a swarm of UAVs. Not only do they learn to track a target, but the robots also learn to communicate better (i.e., the content of the message) for such tracking following the proposed policy gradient technique.

Yasuda and Ohkura [78] used a shared replay memory along with DQN to accelerate the training process for the swarm with regard to path planning. By using more robots contributing their individual experiences to the replay memory, the swarm system was able to learn the joint policy faster. Communication is an important aspect of swarm systems. Usually, researchers use pre-defined communication protocols for coordination among the swarm robots. Hüttenrauch et al. [140] proposed a histogram-based communication protocol for swarm coordination, where the robots use DRL to learn decentralized policies using TRPO [44]. An example task is graph building formation, where the robots aim to cover a certain area through coordination. Another considered task is establishing a communication link between the robots and connecting two points on a map. Along the same line, in 2019, Hüttenrauch et al. [141] used TRPO again to find MADRL-based solutions for rendezvous and pursuit–evasion in a swarm system. The main contribution of their work is the incorporation of Mean Embedding [232] into the DRL method they use to simplify the state information each agent obtains from other agents. Up to 100 robots were used in simulation experiments.

### 3.4. Pursuit-Evasion

In a pursuit–evasion game, usually, multiple pursuers try to capture potentially multiple evaders. When all the evaders are captured or a given maximum time elapses, the game finishes [233,234,235]. For a detailed taxonomy of such problems, the reader is referred to [233]. Some of the sensors that the robots might use in this application include sonar, LiDAR, and 3D cameras, among others. A unified model to analyze data from a suit of sensors can also be used [236]. An illustration is shown in Figure 10.

Egorov [59] proposed a solution for the classic pursuit–evasion problem [233] using an extension of single-agent DQN, called multi-agent DQN (MADQN). The state space is represented as a four-channel image consisting of a map, opponent location(s), ally location(s), and a self-channel. Yu et al. [40] proposed the use of a decentralized training method for pursuit evasion where each agent learns its policy individually and used limited communication with other agents during the training process. This is unlike traditional MADRL techniques where the training is centralized. The execution of the policy for each agent is also decentralized.

Wang et al. [23] proposed to extend a MARL algorithm called cooperative double Q-learning (Co-DQL) for the multi-UAV pursuit–evasion problem. The foundation of Co-DQL is Q-networks with multi-layer perceptrons. Unlike traditional applications where the evader might move around randomly, in this paper, the authors assume that the target also learns to move intelligently up to a certain degree via RL. In [237], the authors consider a setup with one superior evader and multiple pursuers. They use a centralized critic model, where the actors are distributed. Unlike traditional broadcasting techniques, the authors smartly use a leader–follower line topology network for inter-robot communication that reduces the communication cost drastically. Although not strictly pursuit-evasion, Zhang et al. [76] use MADRL for coordinated territory defense, which is modeled as a game where two defender robots coordinate to block an intruder from entering a target zone.

Gupta et al. [81] argue that instead of using a centralized multi-agent DRL framework, where the model learns joint actions from joint states and observations, a more sophisticated parameter-sharing approach can be used. A drawback of the centralized learning system is that the complexity grows exponentially with the number of agents. The authors use TRPO as their base algorithm and the policy is trained with the experiences of all agents simultaneously via parameter sharing. The multi-agent scenarios they use for testing the quality of the proposed solution are pursuit–evasion and a multi-walker system with bipedal walkers.

### 3.5. Information Collection

The objective of information gathering about an ambient phenomenon (e.g., temperature monitoring or weed mapping) using a group of mobile robots is to explore parts of an unknown environment, such that uncertainty about the unseen locations is minimized. Relevant sensors for information gathering include RGB, Normalized Difference Vegetation Index (NDVI), or multi-spectral cameras, and thermal and humidity sensors, among others. This is unlike coverage, where the goal is to visit all the locations. There are two main reasons for this: (1) information (e.g., temperature measurements) in nearby points are highly correlated, and, therefore, the robots do not need to go to all the locations within a neighborhood [238]; and (2) the robot might not have enough battery power to cover the entire environment. This is especially true in precision agriculture, where the fields are usually too large to cover [18]. An illustration is shown in Figure 11, where the robots are tasked with collecting information from their unique sub-regions, and through communication, they will need to learn the underlying model.

Viseras and Garcia [240] have developed a novel DRL algorithm based on the popular A3C [45] algorithm. They also provide a model-based version of their original algorithm for gathering information, which uses CNNs. Said et al. [241] have proposed a mean field-based DRL technique that uses an LSTM module—a type of recurrent neural network for multi-robot information collection about an unknown ambient phenomenon. The robots are battery-powered with limited travel ranges. Recently, Wei and Zheng [67] also used MADRL for multi-robot informative path planning. They develop two strategies for cooperative learning: (1) independent Q-learning with credit assignment [4], and (2) sequential rollout using a GRU. Along the same line, Viseras et al. [85] have proposed using a MADRL framework for a multi-robot team to monitor a wildfire front. The two main components in this framework are (1) individually-trained Q-learning robots and (2) value decomposition networks. The authors have used up to 9 UAVs for testing the efficiency of their presented work.

### 3.6. Task Allocation

Multi-robot task allocation (MRTA) is a combinatorial optimization problem. Given a set of *n* robots and *m* tasks, the goal is to allocate the robots to the tasks such that a given utility function is optimized. Now, if multiple robots need to form a team to complete a single task, then it is a single-task, multi-robot allocation problem. On the other hand, if one robot can offer its services to multiple tasks, then it is called a single-robot, multi-task allocation problem. The robots might be connected to a central server via Wi-Fi, e.g., in a warehouse setting, and can receive information about tasks and other robots. Similarly, communication can happen with other robots via this central server using Wi-Fi as well. Overhead cameras or tracking systems can be used for robot localization in such a scenario. Comprehensive reviews about such MRTA concepts and solutions can be found in [242,243]. An example task allocation scenario is presented in Figure 12.

Elfakharany and Ismail [132] developed a novel multi-robot task allocation and navigation method. This is the first work to propose a MADRL method to tackle task allocation, as well as the navigation problem. They use PPO with actor–critic. Their centralized training and decentralized execution method uses CNNs. Paul et al. [129] proposed to use DRL for multi-robot task allocation. They proposed a neural network architecture that they called a Capsule Attention-based Mechanism, which contains a Graph Capsule Convolutional Neural Network (GCapCN) [244] and a Multi-head Attention mechanism (MHA) [245,246]. The underlying architecture is a GNN. The task graph is encoded using GCapCN and combined with the context, which contains information on the robot, time, and neighboring robots. This information is then decoded with the MHA. Although not strictly task assignment, MADRL has been used for forming teams of heterogeneous agents (such as ambulance and fire brigade in a rescue operation) to complete a given task by Goyal [86]. Goyal has applied this technique for training a team of fire brigades to collaboratively extinguish a fire in a city within the Robocup Rescue Simulator.

Devin et al. [247] developed a novel method of compartmentalizing a trained deep reinforcement learning model into task-specific and robot-specific components. Due to this, the policies can be transferred between robots and/or tasks. Park et al. [114] propose a PPO-based DRL technique for task allocation. Their solution is tested with single-task, multi-robot, and time-extended assignments. They use an encoder–decoder architecture to represent robots and tasks, where a cross-attention layer is used to derive the relative importance of the tasks for the robots.

Scheduling tasks is another important aspect of task planning. Wang and Gombolay [82] used GNNs and imitation learning for a multi-robot system to learn a policy for task scheduling. The proposed model is based on graph attention networks [227]. The scheduling policy is first learned using a Q-network with two fully-connected layers. Imitation learning is then used to train the network from an expert dataset that contains schedules from other solutions. On the other hand, Johnson et al. [93] study the problem of dynamic flexible job shop scheduling, where an assembly line of robots must dynamically change tasks for a new job series over time. The robots learn to coordinate their actions in the assembly line. Agrawal et al. [52] performed a case study on a DRL approach to handling a homogeneous multi-robot system that can communicate while operating in an industry setting. PPO is used as the foundation algorithm. The objective of this work is to train the robots to work with each other to increase throughput and minimize the travel distances to the allocated tasks while taking the current states of the robots and the machines on the floor into account.

One of the most recent studies on deep RL-based MRTA is due to [89], which aims to use DRL for the parallelization of processing tasks for MRTA. The authors base their method on Branching Dueling Q-Network [248] with respect to multi-robot search and rescue tasks. In such a network, multiple branches of a network shares a common decision-making module where each branch handles one action dimension. This helps to reduce the curse of dimensionality in the action space. In total, 20 robots have been used within a simulation to test the feasibility of the proposed technique.

A very different and interesting task assignment application in defense systems is studied by Liu et al. [120]. The authors presented a DRL framework for multi-agent task allocation for weapon target assignment in air defense systems. They use PPO-clip along with a multi-head attention mechanism for task assignments of a (army) general and multiple narrow agents. The neural network architecture uses fully connected layers and a GRU. The major aim of this work is to increase processing efficiency and solution speed of the multi-agent task assignment problem at a large scale. Simulation experiments were carried out in a virtual digital battlefield. The experimental setup includes offensive forces and defensive forces. The defensive forces have places to protect and need to make real-time task allocation decisions for defense purposes. The defensive forces are tested with 12 and the offensive forces are tested with a total of 32 agents.

### 3.7. Object Transportation

To transport an object using two or more cooperative mobile robots, the goal is to design a strategy where the robots’ actions are highly coordinated. Communication among the robots may or may not be possible. The robots can use depth cameras or laser scanners for avoiding obstacles. On the other hand, an optic-flow sensor can be used to determine if the pushing force from the robot has resulted in any object movement or not [249]. A force-torque sensor can be used on the robot to measure the force amount placed on the object. For a comprehensive review of this topic, please refer to [250]. An illustration is shown in Figure 13.

Zhang et al. [73] have used a modified version of DQN that controls each robot individually without a centralized controller or a decision maker. To quantitatively measure how well the robots are working together, they use the absolute error of estimated state–action values. The main idea is to use DQN to have homogeneous robots carry a rod to a target location. Each robot acts independently with neither leading nor following. Niwa et al. [251] proposed a MADRL-based solution to the cooperative transportation and obstacle removal problem. The basis of their solution is to use MARL to train individual robots’ decentralized policies in a virtual environment. The policies are trained using MADDPG [61]. The authors then use the trained policies on real teams of robots to validate the effectiveness. The robots are supposed to push a target object to a final waypoint while moving a physical barrier out of the way to accomplish the task. Manko et al. [77] used CNN-based DRL architecture for multi-robot collaborative transportation where the objective is to carry an object from the start to the goal location. Eoh and Park [79] proposed a curriculum-based deep reinforcement learning method for training robots to cooperatively transport an object. In a curriculum-based RL, past experiences are organized and sorted to improve training efficiency [252]. In this paper, a region-based curriculum starts by training robots in a smaller area, before transitioning to a larger area and a single-robot to multi-robot curriculum begins by training a single robot to move an object, then transferring that learned policy to multiple robots for multi-robot transportation.

### 3.8. Collective Construction

In a collective construction setup, multiple cooperative mobile robots are required. The robots might have heterogeneous properties [253]. The robots can follow simple rules and only rely on local information [254]. In the popular TERMES project from Harvard University [254], a large number of simple robots collect, carry, and place building blocks to develop a user-specified 3D structure. The robots might use onboard vision systems to access the progress in construction. A force sensor-equipped gripper can be used for holding the materials. Furthermore, a distance sensor, e.g., sonar can be used for maintaining a safe distance from the construction as well as other robots [255].

Sartoretti et al. [256] developed a framework using A3C to train robots to coordinate the construction of a user-defined structure. The proposed neural network architecture includes CNNs and an LSTM module. Each robot runs its own copy of the policy without communicating with other agents during testing.

A summary of the state and action spaces and reward functions used in some of the papers reviewed in this article are listed in Table 2.

## 4. Challenges and Discussion

Although we find that a plethora of studies have used multi-agent deep reinforcement learning techniques in recent years, a number of challenges remain before we can expect wide adaptation of them in academia as well as commercially. One of the biggest challenges that we identify is scalability. Most of the papers reviewed in this article do not scale beyond tens of robots. This limits real-world adaptation. Although this is an issue with multi-robot systems in general, the data-hungry nature of most of today’s DRL techniques makes the situation worse. In the future, the research community needs to come up with lightweight techniques that potentially are inspired by nature, such as swarming in biology or particle physics while making necessary changes to the underlying RL technique to fit these appropriately.

The second drawback we found in most of the studies is the lack of resources to make them reproducible. One of the overarching goals of academic research is that researchers across the world should be able to reproduce the results reported in one paper and propose a novel technique that potentially advances the field. In the current setup, most papers employing MADRL use their own (simulation) environments for their robots, which makes it extremely difficult for others to reproduce the results. As a community, we need to come up with an accepted set of benchmarks and/or simulators that the majority of the researchers can use for method design and experiments, which, in turn, will advance the field.

The next challenge is to transfer the learned models to real robots and real-world applications. We find that most experiments in the literature are conducted virtually, i.e., in simulation, rather than with physical robots. This leads to a gap in understanding the feasibility. This corroborates the finding by Liang et al. [257]. Unless we can readily use the learned models on real robots in real-world situations, we might not be able to widely adopt such techniques. It is tied up with the previously-mentioned issue of scalability. Additionally, in the deployment phase, the algorithms need to be lightweight while considering the bandwidth limitation for communication among the robots.

Software plays a significant role in developing and testing novel techniques in any robotic domain and applications of MADRL are no different. Here, we discuss some software that are popularly used for testing the feasibility of the proposed techniques in simulation.

VMAS: Vectorized Multi-Agent Simulator for Collective Robot Learning (VMAS) is an open-source software for multi-robot application benchmarking [258]. Some applications that are part of the software include swarm behaviors, such as flocking and dispersion, as well as object transportation and multi-robot football. Note that it is a 2D physics simulator powered by PyTorch [259].MultiRoboLearn: Similar to VMAS, this is an open-source framework for multi-robot deep reinforcement learning applications [260]. The authors aim to unify the simulation and the real-world experiments with multiple robots via this presented software tool, which is accomplished by integrating ROS into the simulator. Mostly multi-robot navigation scenarios were tested. It would be interesting to extend this software to other multi-robot applications, especially where the robots might be static.MARLlib: Although not strictly built for robots, Multi-Agent RLlib (MARLlib) [261] is a multi-agent DRL software framework that is built upon Ray [262] and its toolkit RLlib [263]. This is a rich open-source software that follows Open AI Gym standards and provides frameworks for not only cooperative tasks, but for competitive multi-robot applications as well. Currently, ten environments are supported by MARLlib among which the grid world environment might be the most relevant one to the multi-robot researchers. Many baseline algorithms including the ones that are highly popular among roboticists, e.g., DDPG, PPO, and TRPO are available as baselines. The authors also show that this software is much more versatile than some of the existing ones including [264,265].

Not only these specialized ones, but other traditional robot simulators, such as Webots [266], V-rep [267], and Gazebo [157], can also be used for training and testing multiple robots. These established software platforms provide close-to-reality simulation models for many popular robot platforms. This is especially useful for robotics researchers as we have seen in this survey that MADRL applications range from aerial and ground robots to underwater robots and manipulators. Table 3 summarizes the main types of robots that have been used for MADRL applications. Not only software development, but another challenge is training data. As most of the state-of-the-art algorithms rely on massive amounts of training data, it is not always easy to train a robot with sufficient data. Dasari et al. [268] have created an open-source database for sharing robotic experiences. It contains 15 million video frames from 7 different robot manipulators including Baxter, Sawyer, Kuka, and Fetch arms. Researchers can use this dataset for efficient training while adding new experiences from their experiments to the dataset itself.

As we have seen, sensors play a major role in creating the perception about the environment, as well as aiding the robots with communication capabilities. The robots might need to collect multi-modal sensor data and fuse them for better perceptions. These sensory observations about the environment can then be used as state inputs to the deep neural networks. Modern sensors have high sampling rates, e.g., standard LiDAR samples over 1 million data points per second. Without state-of-art learning mechanisms, it would have been almost infeasible to process and extract meaningful information from such large amounts of data (for tasks such as target recognition, classification, and semantic feature analysis, among others).

Although many robotic applications are utilizing the progress in multi-agent reinforcement learning, we have not seen any paper on modular self-reconfigurable robotics (MSRs) [270,271,272,273] where MADRL has been utilized. We believe that the field of modular robots can benefit from these developments especially given the fact that MSRs can change their shapes and the new shape might not have been pre-defined. Therefore, its control is undefined as well and it might need to learn to move around and complete tasks on-the-fly using techniques, such as MADRL, where each module acts as an intelligent agent.

On the other hand, we have found MADRL-based solutions for manipulation and motion separately. The next question that should be answered is how one can simultaneously learn those two actions where they might affect each other. For example, in a scenario, where multiple UAVs are learning to maintain a formation while manipulating an object with their onboard manipulators. This task would potentially require the robots to learn two actions simultaneously. The research question then would be how to best model the agents, their goals, and the rewards in this complex scenario.

## 5. Conclusions

In this paper, we have reviewed state-of-the-art studies that use multi-agent deep reinforcement learning techniques for multi-robot system applications. The types of such applications range from exploration and path planning to manipulation and object transportation. The types of robots that have been used encompass ground, aerial, and underwater applications. Although most applications involve mobile robots, we reviewed a few papers that use non-mobile (manipulator) robots as well. Most of the reviewed papers have used convolutional neural networks, potentially combining them with fully connected layers, recurrent layers, and/or graph neural networks. It is worth investigating such reinforcement learning techniques for robotics as they have the potential to learn high-level causal relationships among the robots, as well as between the robots and their environment, which might have been extremely difficult to model using a non-learning approach. As better hardware is available on a smaller scale and at a lower price, we expect to see significant growth in novel multi-robot system applications that use multi-agent reinforcement learning techniques. Furthermore, with the progress of the field of artificial intelligence in general, we expect that more studies will have theoretical underpinnings along with their showcased empirical advancements. Although a number of challenges remain to be solved, we are perhaps not too far away from seeing autonomous robots tightly integrated into our daily lives.

## Figures and Tables

**Figure 1 sensors-23-03625-f001:**
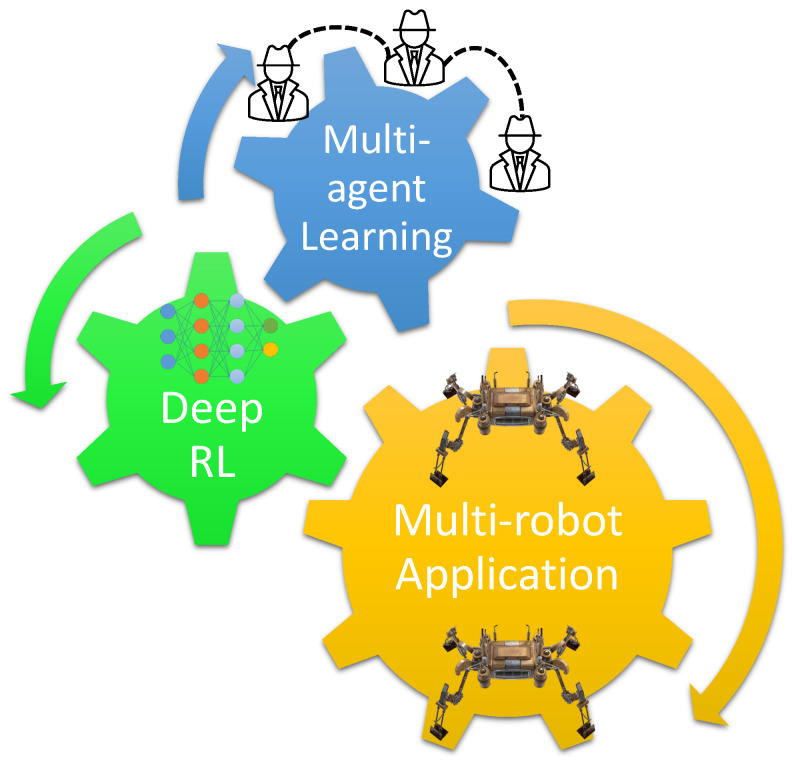
The main contribution of this article is that we have reviewed the latest multi-robot application papers that use multi-agent learning techniques via deep reinforcement learning. Readers will be able to find out how these three concepts are used together in the discussed studies and this survey will provide them with insight into possible future developments in this field, which, in turn, will advance the state-of-the-art.

**Figure 2 sensors-23-03625-f002:**
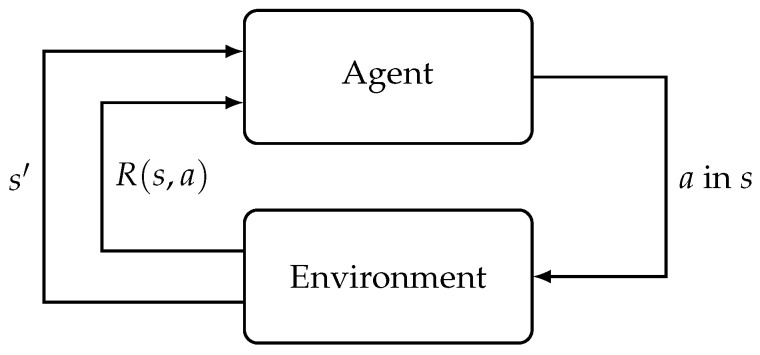
Illustration of Reinforcement Learning.

**Figure 3 sensors-23-03625-f003:**
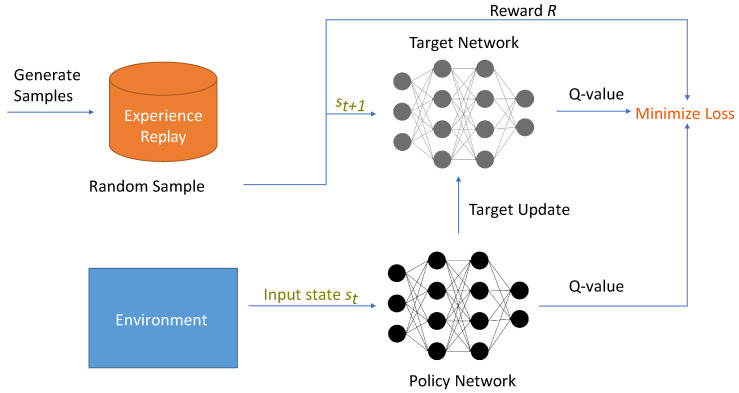
An illustration of the DQN architecture with a target network and an experience replay.

**Figure 4 sensors-23-03625-f004:**
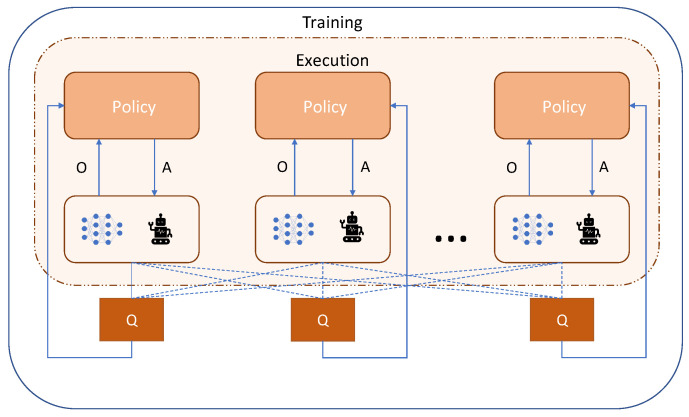
Illustration of multi-agent DDPG (MADDPG) [61].

**Figure 5 sensors-23-03625-f005:**
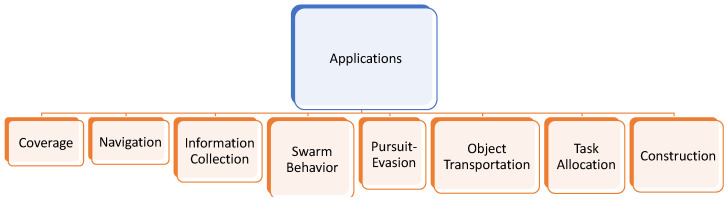
Multi-robot system applications that primarily use MADRL techniques.

**Figure 6 sensors-23-03625-f006:**
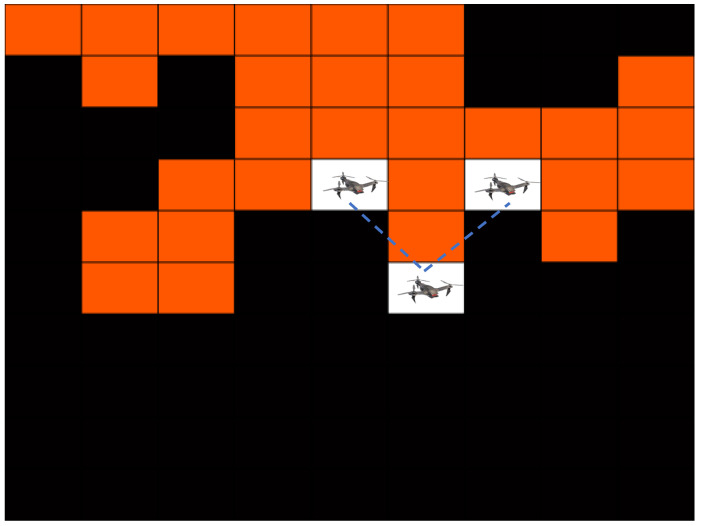
A multi-robot coverage scenario under continuous connectivity: the black, orange, and white cells represent the unvisited, previously visited, and the current cells of robots, respectively. The dotted lines represent the available communication channels among the robots. The goal is to cover the entire environment while maintaining a connected communication network throughout the mission.

**Figure 7 sensors-23-03625-f007:**
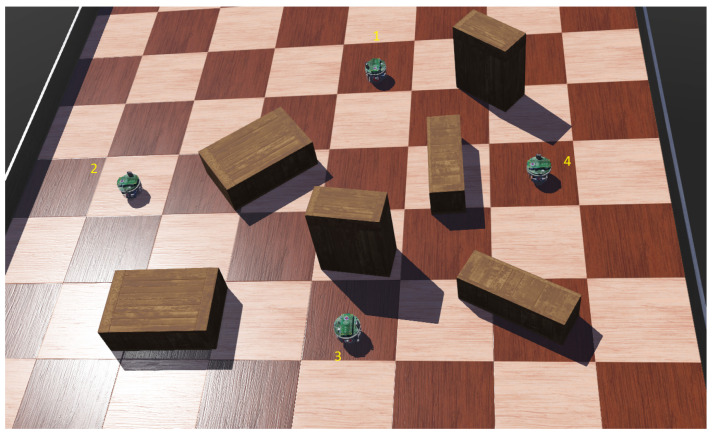
An illustration of multi-robot path planning with 4 e-puck robots, where the starting positions are shown and each robot’s goal position is its orthogonal robot’s starting location. The boxes represent the obstacles in the environment.

**Figure 8 sensors-23-03625-f008:**
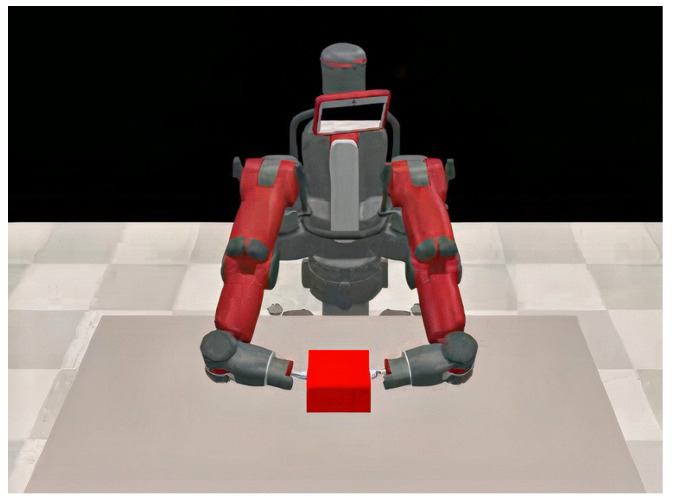
An illustration of dual-arm manipulation is shown using a Baxter robot where each arm might act as an RL agent.

**Figure 9 sensors-23-03625-f009:**
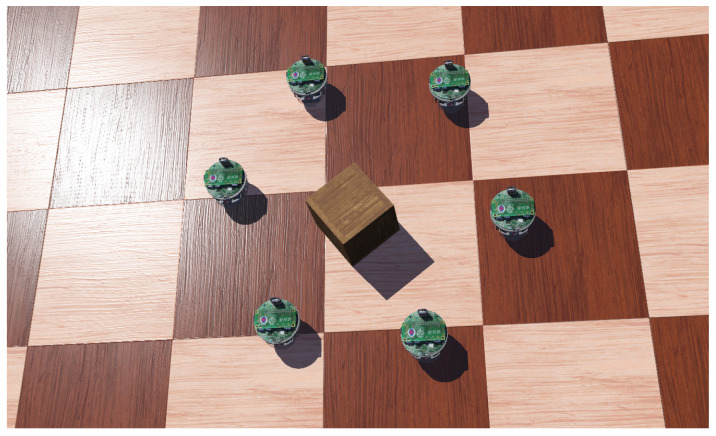
An illustration of pattern (circle) formation with five e-puck robots which are guarding a box in the center.

**Figure 10 sensors-23-03625-f010:**
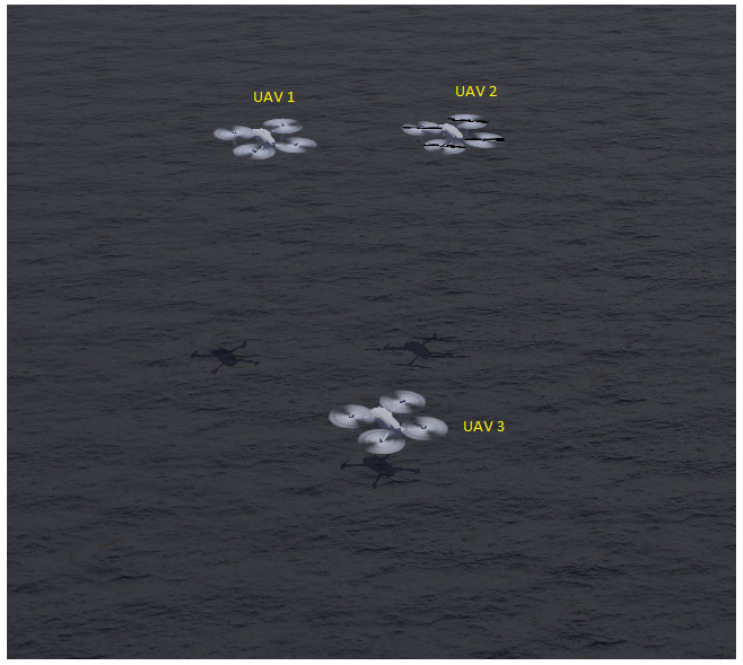
An illustration of multi-robot pursuit–evasion scenario where UAVs 1 and 2 have captured and “grounded” UAV 3, which is an evader robot.

**Figure 11 sensors-23-03625-f011:**
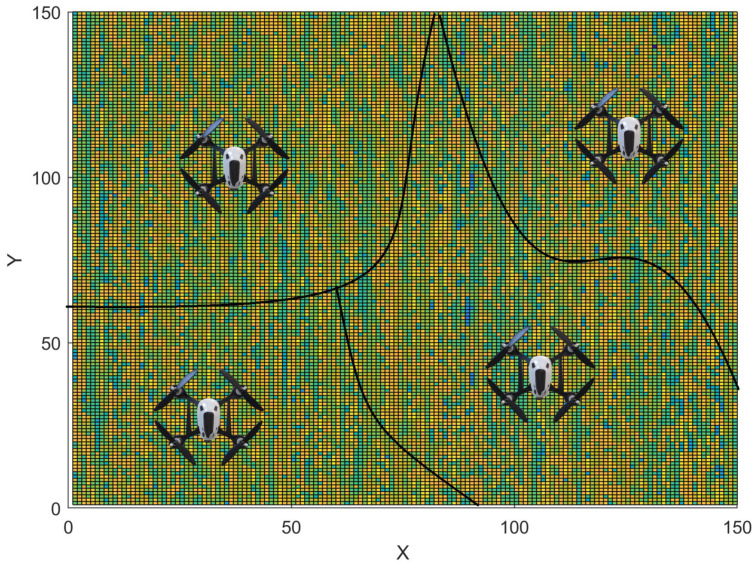
An illustration of multi-robot information collection. The environment is divided into four sub-regions and the underlying heatmap represents measures of soil acidity in parts of the USA [239].

**Figure 12 sensors-23-03625-f012:**
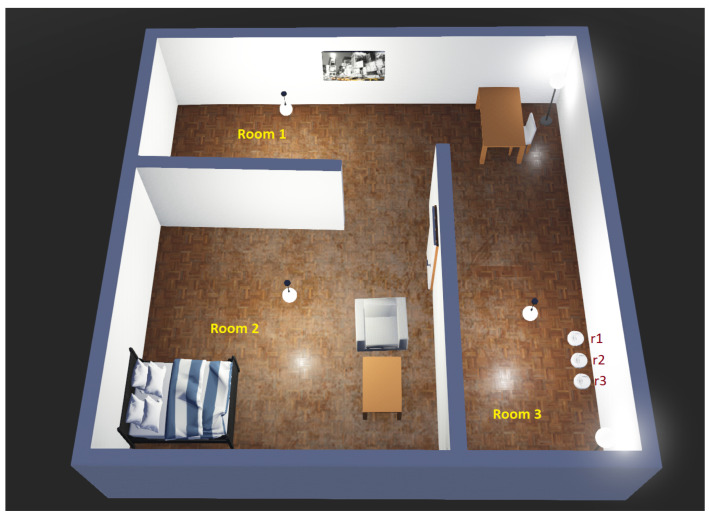
An illustration of multi-robot task allocation: there are 3 iRobot Roombas (r1–r3) and 3 rooms to clean. In a one-to-one matching scenario, the objective would be to assign one Roomba to a certain room. However, as room 2 is larger in size, two robots might be needed to clean it, whereas the third robot (r3) might be assigned to rooms 1 and 3.

**Figure 13 sensors-23-03625-f013:**
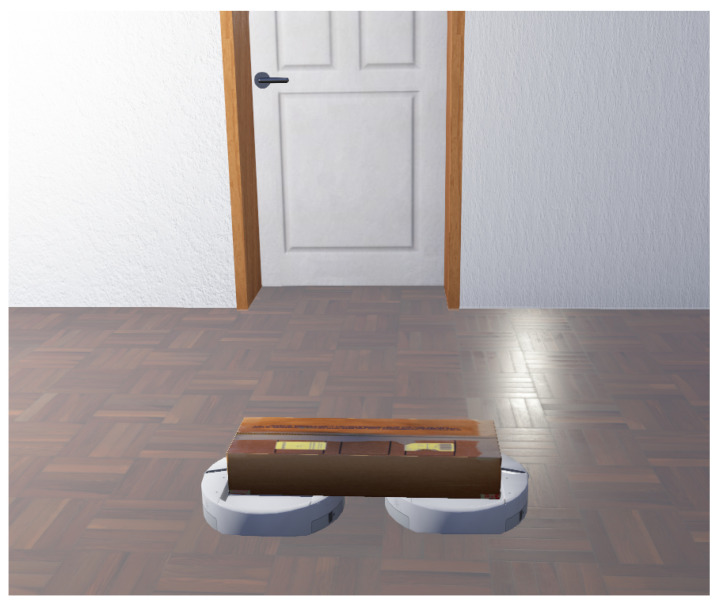
An illustration of multi-robot object transportation is presented where 2 iRobot Create robots carry a cardboard box and plan to go through a door in front of them.

**Table 1 sensors-23-03625-t001:** Types of deep RL algorithms used in the surveyed papers are listed. If a popular algorithm is used as a foundation, the algorithm’s name is also mentioned within parentheses.

Q-Networks	Policy Gradients		
	**DDPG**	**PPO**	**Other**
[23,24,39,40,59,67,68,69,70,71,72,73,74,75,76,77,78,79,80,81,82,83,84,85,86,87,88,89,90,91,92] (QMIX), [38] (DDQN), [93] (DDQN), [94] (DDQN), [95] (DQN), [96] (DQN)	[46,47,48,49,50,97,98,99,100,101,102,103,104,105,106],	[22,52,53,54,55,56,57,107,108,109,110,111,112,113,114,115,116,117,118,119,120,121,122,123,124,125,126,127,128]	[86,88,129,130,131,132,133,134,135,136,137,138,139,140] (TRPO), [141] (TRPO), [81] (TRPO), [142] (TD3), [143] (SAC), [144] (SAC)

**Table 2 sensors-23-03625-t002:** Examples of state and action spaces and reward functions used in prior studies.

Refs.	State	Action	Reward
[59]	Map of the environment and robots’ locations with 4 channels	Discreet	Based on the locations of the robots
[67]	Robot locations and budgets	Discreet	Based on collected sensor data
[68]	Position of the leader UAVs, the coverage map, and the connection network	Discreet	Based on the overall coverage and connectivity of the Leader UAVs.
[69]	Map with obstacles and the coverage area	Discreet	Based on the robot reaching a coverage region within its task area.
[24]	Map with covered area	Discreet	Based on robot coverage.
[71]	Map of the environment	Discreet	Based on the robot movements and reaching the target without collisions.
[70]	Map with robots’ positions	Discreet	Based on the herding pattern.
[72]	Map with robots’ positions and target locations	Discreet	Distance from the goal and collision status.
[39]	Map with robots’ positions and target locations	Discreet	Distance from the goal and collision status.
[40]	Pursuer and evader positions	Discreet	Collision status and time to capture the predator.
[73]	The map, locations, and orientations of the robots, and the objects the robots are connected to	Discreet	Based on the position of the object and the robots hitting the boundaries.
[74]	The map, and the locations of the robots	Discreet	Based on distance from the target and collisions.
[76]	The regions of the robots, positions of the defender and the attacker UAVs and the intruder	Discreet	Based on distance.
[77]	The distance from the MRS center to the goal, the difference in orientation of the direction of MRS to the goal, and the distance between the robots	Discreet	Based on the distance to the goal, orientation to the goal, proximity of obstacles, and the distance between the robots.
[78]	Sensor input information that includes distance to other robots and the target landmarks	Discreet	Based on becoming closer to the target landmark.
[79]	Spatial information on the robots, the object, and the goal	Discreet	Based on the object reaching the goal while avoiding collisions.
[80]	Robot position and velocity	Discreet	Based on the robots being within sensing range of one another.
[38]	The positions and speed of the first responders and UAVs	Discreet	Based on the Cramér–Rao lower bound (CRLB) for the whole system.
[93]	The agents’ positions, types, and remaining jobs	Discreet	Based on minimizing the makespan.
[83]	The position and direction of the leader and the followers	Discreet	Based on the distance from followers to leaders and collision status.
[84]	Information on the target, other agents, maps, and collisions	Discreet	Based on finding targets and avoiding obstacles.
[85]	The robot’s position, position relative to other robots, and angle and direction of the robots	Discreet	Based on covering a location on fire.
[23]	The positions, velocities, and distances between UAVs	Discreet	Determined by the distance from the target and the evader being reached by the pursuer.
[87]	Consists of static obstacle locations and the locations of other agents	Discreet	Based on the robots’ movements toward the goal while avoiding collisions.
[94]	Contains the sensor data for the location of the target relative to the robot and the last action done by the robot	Discreet	Determined by the robot reaching the goal, reducing the number of direction changes, and avoiding collisions.
[95]	A map that includes the agent’s locations, empty cells, obstacle cells, and the location of the tasks	Discreet	Determined by laying pieces of flooring in the installation area.
[110]	Map of the environment represented with waypoints, locations of the UAVs, and points of interest	Discreet	Based on the coverage of the team of robots.
[114]	The positions and tasks of the robots, the state of the robot	Discreet	Based on minimizing the number of timesteps in an episode.
[96]	Map of the area to be sterilized and the positions of the agents, the cleaning priority, size, and area of the cleaning zone	Discreet	Based on the agents cleaning priority areas for sanitation.
[86]	Temperature and “fieryness” of a building, location of the robots, water in the tanks, and busy or idle status	Discreet	Based on keeping the fires to a minimum “fieryness” level.
[53]	Sensory information on obstacles	Discreet	Based on the UAV’s coverage of the area.
[52]	Robots’ positions and velocities and the machine status	Discreet	Determined by robots completing machine jobs to meet the throughput goal, and their motions while avoiding collisions.
[107]	Includes the laser readings of the robots, the goal position, and the robot’s velocity	Continuous	Based on the smooth movements of the robots while avoiding collisions.
[22]	Includes the laser readings of the robots, the goal position, and the robot’s velocity	Continuous	Based on the time to reach the target while avoiding collisions.
[108]	An environment that includes the coordinates of the manipulator arm gripper	Continuous	Based on reaching the target object.
[109]	Laser measurements of the robots and their velocities	Continuous	Based on the centroid of the robot team reaching the goal.
[117]	The state of the robot, other robots, obstacles, and the target position	Continuous	Based on the robots’ relative distance from the target location.
[97]	Sensed LiDAR data	Continuous	Based on the robot approaching and arriving at the target, avoiding collisions and the formation of the robots.
[132]	Consists of the goal positions, the robots’ positions, past observations	Continuous	Based on the robot moving towards the goal in the shortest amount of time.
[133]	Contains laser data, speeds and positions of the robots, and the target position	Continuous	Based on arriving at the target, avoiding collisions, and relative position to other robots.
[113]	The position information of other robots (three consecutive frames)	Continuous	Based on time for formation, collisions, and the formation progress.
[115]	The most recent three frames of the map, local goals that include positions and directions	Continuous	Based on minimizing the arrival time of each robot while avoiding collisions.
[116]	Map of the environment, robot positions and velocities, and laser scans	Continuous	Based on the arrival time of the robot to the destination, avoiding collisions, and smoothness of travel.
[104]	The coverage score and coverage state for each point of interest and the energy consumption of each UAV	Continuous	Defined by coverage score, connectivity fairness, and energy consumption.
[134]	Robot’s relative position to the goal and its velocity	Continuous	Based on the robots having collisions.
[88]	Robot motion parameters, relative distance and orientation to the goal, and their laser scanner data	Discreet/Continuous	Determined by reaching the goal without timing out and avoiding collisions.

**Table 3 sensors-23-03625-t003:** Types of robots in the reviewed papers. If the type is not specified in the paper, it is not listed here.

Aerial Robots	Ground Robots	Manipulators
[23,24,38,46,53,56,57,68,76,83,85,91,104,106,110,120,123,134,135,138,139,141,142,161,167,171,172,186,205,207,218,219,220,230,240,269]	[22,39,49,52,54,55,59,70,71,73,74,75,77,78,79,82,87,90,92,97,107,109,111,112,115,117,120,124,128,130,132,133,137,155,162,183,189,190,194,202,212,251]	[50,93,108,131,143,144]

## Data Availability

Not applicable.

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
