# Peer review of "Multi-Agent Deep Reinforcement Learning for Multi-Robot Applications: A Survey"

_sensors, 2023, doi:10.3390/s23073625_

Round 1

Reviewer 1 Report

This paper reviews several multi-agent deep reinforcement learning methods and applications where multiple agents present in the environment not only learn from their own experiences but also from each other and its applications in multi-robot systems. The Paper reviews several applications such as coverage, navigation, transportation, among Others. This work also make a good discussion related to current and future challenges of the multi-agent deep reinforcement learning methods. My comments to the authors are listed below.

The Paper is very well structured. The introduction is short, but the theme is also quite new.

- Positive things:

It is appreciated that the authors make an introduction to what are the learning algorithms for reinforcement before making the literature review.

The graphs of the article are very good.

The bibliography of the article is correct.

The topic is very interesting, updated, and high impact.

- Things that the article must improve:

Although the review of the article in general is adequate, there are some pending doubts since, for example, the motivation of each author is not understood very well in the selection of a determined algorithm for its application. The selection of the reinforcement learning algorithm is usually linked to the type of actions in the MPD. For example, if the actions are discrete, you can use Deep Q-learning, but if they are continuous, this algorithm no longer usually gives good results. However, PPO usually works well with actions that are continuous and discrete. I recommend that the authors improve table 1 or use an additional table to indicate for each method if the actions are continuous or discrete. Likewise, it would be interesting to have information on how actions, states, and reward for each case were modeled. This would make the contribution of this article extraordinary.

Example:

If the Paper makes this improvement, it would be ready to be published

Author Response

Although the review of the article in general is adequate, there are some pending doubts since, for example, the motivation of each author is not understood very well in the selection of a determined algorithm for its application. The selection of the reinforcement learning algorithm is usually linked to the type of actions in the MPD. For example, if the actions are discrete, you can use Deep Q-learning, but if they are continuous, this algorithm no longer usually gives good results. However, PPO usually works well with actions that are continuous and discrete. I recommend that the authors improve table 1 or use an additional table to indicate for each method if the actions are continuous or discrete. Likewise, it would be interesting to have information on how actions, states, and reward for each case were modeled. This would make the contribution of this article extraordinary.

Rebuttal:

We agree and we have added a table (2) to summarize the state, action, and rewards of some of the most interesting papers listed in our survey. We have not reproduced the table here for brevity.

We appreciate this feedback.

Reviewer 2 Report

The article reported a review on the the use of multi-agent deep reinforcement learning on multi robot applications. The review includes about 266 references of which are related to the topics and up-to date. However, the subject matter is not in the scope of the journal thus I am unable to recommend the article for publication.

These are some of my suggestion on improvement of the article:

1) The abstract and the introduction should include on the use of deep reinforcement learning for sensory based applications/functions of multi robots.

2) Pg2 P2 the paragraph should mentioned the sensory capabilities of multi agent deep reinforcement learning.

3) Subsection 3: this section do mentioned on the use of the technique a part of sensory algorithms.

4) Subsection 3.2: this section to be rewritten on the use of the algorithm in sensory system/ environmental scanning for multi robot system.

5) Subsection 3.3, 3.8: this is not in the scope of the journal

6) Subsection 3.4, 3.5, 3.6: this section to be rewritten on the use of the algorithm in sensory system/ environmental scanning for behavior modeling/pursuit/task allocation.

7) Subsection 4: The challenges in the discussion to include sensory based challenges

8) The conclusion must be in accordance to the scope of the journal.

Author Response

The article reported a review on the the use of multi-agent deep reinforcement learning on multi robot applications. The review includes about 266 references of which are related to the topics and up-to date. However, the subject matter is not in the scope of the journal thus I am unable to recommend the article for publication.

Rebuttal:

We understand your concern and value your opinion.

We would like to mention two important factors here. First, although the journal is Sensors, this article is submitted to a special issue named “Mobile Robots: Navigation, Control and Sensing” (https://www.mdpi.com/journal/sensors/special_issues/MobileRobots).  This special issue covers several topics discussed in our survey including 1) path planning, 2) adaptive robot control, 3) tracking, 4) robot formation and 5) map-based systems, among others. Furthermore, we talked to the editors before submitting our article and we only submitted our article after we have received the green light from them.

The abstract and the introduction should include on the use of deep reinforcement learning for sensory based applications/functions of multi robots. Pg2 P2 the paragraph should mentioned the sensory capabilities of multi agent deep reinforcement learning.

Rebuttal:

Although this paper is not specifically about sensor systems, to accommodate your comments, we have added the following text to the Introduction section.

Robots' onboard sensors play a significant role in such applications. For example, the state space of the robots might include the current discovered map of the environment, which could be created by the robots' laser scanners~\cite{long2018towards}. The state might also include locations and velocities, for which the robot might need sensory information from GPS or an overhead camera~\cite{wang2020cooperatively}. Furthermore, vision systems, such as regular or multi-spectral cameras can be used by the robots to observe the current state of the environment, and data collected by such cameras can be used for robot-to-robot coordination~\cite{pham2018cooperative}.

Subsection 3: this section do mentioned on the use of the technique a part of sensory algorithms. Subsection 3.2: this section to be rewritten on the use of the algorithm in sensory system/ environmental scanning for multi robot system.

Rebuttal

 We are not sure about what you meant in this comment. In Section 3, we have talked about different multi-robot applications. Furthermore, in Table 2 now, we have listed the state information of the robots. As can be seen that robots’ perception of the environment is coming from their sensory information. For example, a map can be built from laser scans, an information field can be created from the onboard information collection sensor such as a humidity sensor. On the other hand, robots' positions on the map and their relative distances from the target location can be calculated from GPS data. However, these are fed into the learning system, which is our focus in the paper, i.e., how this input sensory data is being used to learn something meaningful, and not the hardware sensor system itself.

Subsection 3.3, 3.8: this is not in the scope of the journal

Rebuttal

As mentioned previously, this article was submitted to a special issue in Sensors that concentrates on navigation, control, and sensing. Swarm behavior (section 3.3) modeling involves topics such as flocking, pattern formation, and tracking. All of these topics require navigation, i.e., movements of the robots, and control mechanisms, while sensed data is being used as robot perceptions. Similarly, construction (section 3.8) involved robot motion, control of manipulators, and sensing to decide how much construction is complete and how much is still left. Therefore, in our humble opinion, both these sections are of utmost relevance to the special issue topics.

Subsection 3.4, 3.5, 3.6: this section to be rewritten on the use of the algorithm in sensory system/ environmental scanning for behavior modeling/pursuit/task allocation.

Rebuttal

Unfortunately, our paper is not about sensory systems – it is about multi-robot applications. The sensors are merely used for robot perception (e.g., map of the environment via laser scanning and GPS) or communication (robot-to-robot communication via WiFi). Furthermore, the special issue to which we have submitted this article does not require us to specifically talk about this. With all due respect, we believe that adding more text on sensory systems will change the aim/tone of the paper, which we do not agree with. Therefore, respectfully, we will not be able to make this change.

Subsection 4: The challenges in the discussion to include sensory based challenges

Rebuttal

Unfortunately, we could not identify sensor-based challenges for multi-robot applications via MADRL techniques. The challenges are peripherally related to the sensor systems but these MADRL techniques that we have discussed in this paper are not specifically designed for the improvement of sensors, instead, they are for multi-robot coordination and consequent completion of the given task with a group of robots. For this purpose, they used the data from sensors but did not incorporate them in any other way.

The conclusion must be in accordance to the scope of the journal.

Rebuttal

With all due respect, we believe it is within the scope of the journal's special issue to which we have submitted the article.

Reviewer 3 Report

1. The authors may consider including a visual overview of the paper in the introduction to improve the reader's understanding of the article.

2. The authors should clearly state their main contributions at the end of the introduction to distinguish their work from other surveys in the field.

3. In Section 2, it takes up too much space to introduce DRL algorithms. It can be shortened with highlights remaining. 

4. The authors may consider revising Table 1 to categorize algorithms in a clearer and more organized way.

5. In Section 3, the authors should provide references for the figures and cite them in their titles. Also, Fig. 7 is too fuzzy.

Author Response

The authors may consider including a visual overview of the paper in the introduction to improve the reader's understanding of the article.

Rebuttal:

Thank you for the suggestion. We have provided such a picture in Fig. 1. The caption of the figure reads as follows.

The main contribution of this article is illustrated: We have reviewed the latest multi-robot application papers that use multi-agent learning techniques via deep reinforcement learning. A reader would find out how these three concepts are used together in the discussed studies and that will provide them with insight into possible future developments in this field, which in turn will advance the state-of-the-art.

The authors should clearly state their main contributions at the end of the introduction to distinguish their work from other surveys in the field.

Rebuttal:

To the best of our knowledge, this is the ONLY survey that bridges the two areas: multi-robot systems and multi-agent deep reinforcement learning. We have stated this in the Introduction section as well as you can find the statement below for your reference.

The primary contribution of this article is that to the best of our knowledge, this is the only study that surveys the multi-robot applications via multi-agent deep reinforcement learning technologies. This survey gives a foundation for future researchers to read and build upon to develop state-of-the-art multi-robot solutions for applications ranging from task allocation and swarm behavior modeling to path planning and object transportation.

In Section 2, it takes up too much space to introduce DRL algorithms. It can be shortened with highlights remaining. 

Rebuttal:

We appreciate this comment. However, with all due respect, we believe that this view is subjective. For example, Reviewer 1 commended us for such an elaborate introduction. Therefore, we would respectfully like to maintain the current structure of the Introduction section.

The authors may consider revising Table 1 to categorize algorithms in a clearer and more organized way.

Rebuttal:

We have categorized Table 1 in a better way now. The policy gradient algorithm extensions are now subdivided into 3 sub-columns of DDPG, PPO, and others. DDPG and PPO are the most popular policy gradient algorithms in multi-robot applications, so they have their own columns. We appreciate your feedback on this.

In Section 3, the authors should provide references for the figures and cite them in their titles. Also, Fig. 7 is too fuzzy.

Rebuttal:

Note that all the illustrative figures in the paper are created by us (the authors of this article) from scratch. Therefore, we did not cite any previous work.

We have also updated Fig. 7 – it now has a higher resolution. Thank you for this comment.

Round 2

Reviewer 1 Report

Dear editor, the authors have reviewed and corrected the article based on all the suggestions made in the previous revision. In this state, I recommend this article for publication.

Author Response

Thank you for your recommendation.

Reviewer 2 Report

The article reported on the use of multi-agent deep reinforcement leaning on multi robot applications. The review in my opinion is novel and up-to-date on which currently there are no other similar comprehensive review on this matter.  The article is for a special issue of sensors specifically on mobile robots. 

Based on the last review the authors have added few sentences on the role of sensor system and for introduction I find it sufficient.

However, the author do not make any improvements in other areas as I suggested. I expect the review to mentioned how multi-agent deep reinforcement learning make use / enhance the sensor capability of the agent in multi robot applications or on how multi-agent deep reinforcement learning enhances the mobile robots understanding of the environment for completing the tasks.

Author Response

Based on the last review the authors have added few sentences on the role of sensor system and for introduction I find it sufficient.

 Rebuttal:

             Thank you!

However, the author do not make any improvements in other areas as I suggested. I expect the review to mentioned how multi-agent deep reinforcement learning make use / enhance the sensor capability of the agent in multi robot applications or on how multi-agent deep reinforcement learning enhances the mobile robots understanding of the environment for completing the tasks.

Rebuttal:

              For each multi-robot application, we have now added a description of what sensors might be required to accomplish them along with new references. Furthermore, we have added a paragraph on sensor data and learning in Section 4 (Challenges and Discussion). We hope this is what you intended to see.

All the additions to the paper are in red color.

Thank you!